# MGMT Promoter Methylation Prediction in Glioblastoma Using 3D CNNs with Advanced MRI Sequences

**Tran Nguyen Tuan Minh**[1,2]  ID                    TRANNGUYENTUANMINH@GMAIL.COM
[1] *AIBioMed Research Group, Taipei Medical University, Taipei 110, Taiwan*
[2] *Neurology - Stroke - Cardiology Center, Vinmec Da Nang International Hospital, Da Nang, Vietnam*

**Quang Hien Kha**[1,3]                                   D142111015@TMU.EDU.TW
[3] *International Ph.D. Program in Medicine, College of Medicine, Taipei Medical University, Taipei 110, Taiwan*

**Viet Huan Le**[4]                                      HUAN.RESIDENT.DR@GMAIL.COM
[4] *Department of Thoracic Surgery, Khanh Hoa General Hospital, Nha Trang City 65000, Vietnam*

**Matthew Chin Heng Chua**[5]                                MATTHEWCHUA@NUS.EDU.SG
[5] *Department of Biomedical Informatics, Yong Loo Lin School of Medicine, National University of Singapore, Singapore 119228, Singapore*

**Nguyen Quoc Khanh Le**[*1,6]                              KHANHLEE@TMU.EDU.TW
[6] *In-Service Master Program in Artificial Intelligence in Medicine, College of Medicine, Taipei Medical University, Taipei 110, Taiwan*

**Editors:** Accepted for publication at MIDL 2026

## Abstract

Accurate determination of O6-methylguanine-DNA methyltransferase (MGMT) promoter methylation status is essential for therapeutic planning in glioblastoma (GBM). Although molecular assays remain the reference standard, they are costly, invasive, and not always feasible in routine practice. This has motivated the development of non-invasive MRI-based deep learning approaches, particularly those leveraging advanced physiological imaging sequences. In this study, we investigated whether arterial spin labeling (ASL) and apparent diffusion coefficient (ADC) imaging provide complementary information for predicting MGMT methylation status in IDH-wildtype GBM. We analyzed 351 patients from the UCSF Preoperative Diffuse Glioma MRI dataset and trained 3D convolutional neural network models based on a ResNet-10 architecture using ASL, ADC, diffusion-weighted imaging (DWI), and conventional T2-FLAIR sequences. Among single-sequence models, ASL achieved the highest performance (accuracy of 0.76, precision of 0.75, and F1 score of 0.73). A dual-sequence model combining ASL and ADC further improved prediction, yielding an AUC of 0.83, significantly outperforming both the ASL-only model and the T2-FLAIR model (AUC 0.6524; DeLong test, $p < 0.05$). These results demonstrate that integrating perfusion- and diffusion-based MRI captures complementary physiological characteristics relevant to MGMT methylation, offering a more accurate and fully non-invasive alternative for biomarker assessment. Incorporating advanced MRI sequences into deep learning pipelines may support more informed treatment planning and improve clinical decision-making for patients with GBM.

**Keywords:** glioblastoma, MGMT methylation, deep learning, arterial spin labeling, apparent diffusion coefficient, MRI, non-invasive diagnosis

* Corresponding author

## 1. Introduction

Gliomas are the most common primary malignant brain tumors, accounting for nearly 80% of cases (Ostrom et al., 2014; Hanif et al., 2017). Arising from glial cells that support neuronal function (Davis, 2018), these tumors are classified into four grades of increasing aggressiveness according to the World Health Organization (WHO) (Louis et al., 2021). Grade IV gliomas, or glioblastomas (GBM), exhibit rapid infiltration, poor therapeutic response, and high recurrence rates, contributing to their exceptionally poor prognosis (Ghosh et al., 2017; Stupp et al., 2014; Rock et al., 2012).

A growing emphasis in GBM management is the stratification of tumors using molecular biomarkers such as isocitrate dehydrogenase (IDH) mutation status and O6-methylguanine-DNA methyltransferase (MGMT) promoter methylation (Sahm et al., 2023; Sejda et al., 2022; Thomas, 2023; Horbinski et al., 2022). IDH-wildtype gliomas typically show more aggressive biology and worse outcomes (Weller et al., 2015). MGMT promoter methylation, in particular, has substantial therapeutic significance: it predicts sensitivity to alkylating agents such as temozolomide and guides adjuvant treatment decisions (Sonoda et al., 2010; Hegi et al., 2005). However, current molecular assays for assessing MGMT status are invasive, costly, and require specialized laboratory workflows, limiting their availability in many clinical settings (Nguyen et al., 2021; Quillien et al., 2012).

These limitations have motivated efforts to develop non-invasive imaging-based alternatives. Magnetic resonance imaging (MRI) provides rich structural and physiological information, and advanced MRI techniques (including diffusion-weighted imaging (DWI), apparent diffusion coefficient (ADC), and arterial spin labeling (ASL)) offer measurements of microstructural integrity, water diffusivity, and perfusion, respectively (Hu et al., 2024; Shukla et al., 2017). Such physiological markers may reflect tumor heterogeneity associated with MGMT methylation, making them promising imaging surrogates.

Recent deep learning advances have demonstrated the potential of MRI-based models to predict GBM molecular features (Hu et al., 2024; Yogananda et al., 2021; Faghani et al., 2023). While prior studies have primarily relied on conventional sequences (e.g., T2-weighted and FLAIR), emerging evidence suggests that advanced MRI sequences may capture complementary physiological signatures that improve prediction accuracy (Faghani et al., 2023; Kanas et al., 2017; Han et al., 2018; Saxena et al., 2023). Conventional sequences are effective for visualizing tumor boundaries and edema but often lack the sensitivity to functional and molecular characteristics required for robust biomarker prediction.

To address these limitations, this study investigates whether combining advanced perfusion- and diffusion-based MRI modalities enhances the non-invasive prediction of MGMT promoter methylation status in IDH-wildtype glioblastoma. Using a retrospective MRI dataset, we develop and compare several 3D convolutional neural network (CNN) models trained on individual and combined MRI sequences, with a particular focus on ASL and ADC imaging. Rather than proposing a novel network architecture, our objective is to systematically evaluate whether integrating complementary physiological information from perfusion- and diffusion-sensitive MRI can improve predictive performance beyond conventional structural imaging alone.

## 2. Materials and Methods

An overview of the study workflow is shown in Figure 1. The analysis began with MRI data from 351 IDH-wildtype GBM patients. Each scan, originally provided in Neuroimaging Informatics Technology Initiative (NIfTI) format, was converted to Digital Imaging and Communications in Medicine (DICOM) for standardized handling across preprocessing and modeling steps. The dataset was split into a training cohort (n = 280) and a held-out testing cohort (n = 71). For each patient, the central 100 axial slices (corresponding to the region most likely to contain the tumor) were extracted and processed through a standardized pipeline comprising resizing, augmentation, and normalization. Separate 3D CNN models were trained using individual MRI sequences (ASL, ADC, and DWI), and a dual-sequence fusion model was constructed by combining ASL and ADC inputs along the channel dimension. A comparative model based on conventional T2 and FLAIR sequences was also evaluated. Model performance was assessed on the independent test set.

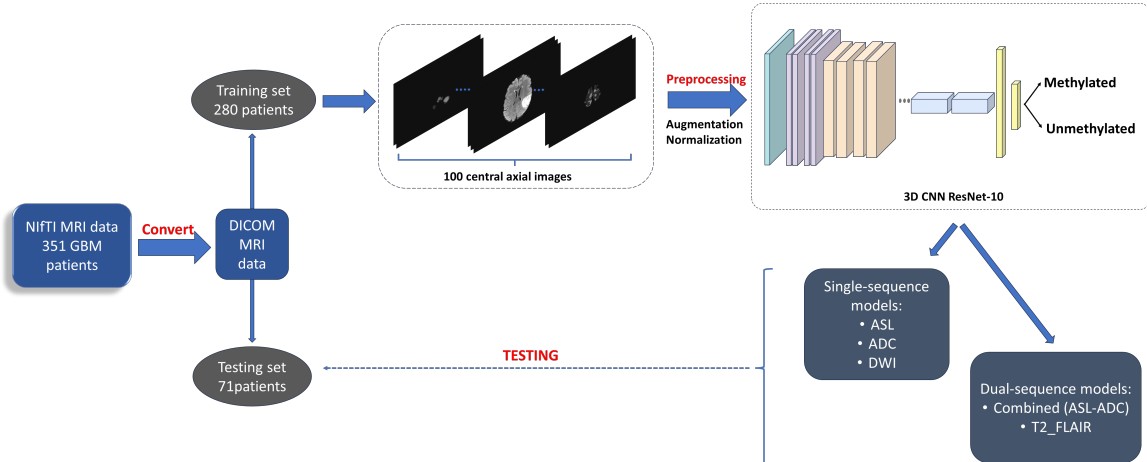

Figure 1: Overview of the proposed workflow for MRI-based MGMT methylation prediction. The pipeline includes MRI acquisition, NIfTI-to-DICOM conversion, slice extraction, preprocessing, and model development using 3D convolutional neural networks (3D CNNs) based on a ResNet-10 architecture. ASL, ADC, DWI, and conventional sequences are processed individually or in combination to generate prediction outputs. Abbreviations: CNN = Convolutional Neural Network; ResNet = Residual Network; NIfTI = Neuroimaging Informatics Technology Initiative; DICOM = Digital Imaging and Communications in Medicine.

### 2.1. Patient data

This study utilized the University of California, San Francisco Preoperative Diffuse Glioma MRI (UCSF-PDGM) dataset (Calabrese et al., 2022), which comprises imaging and molecular profiling data from 501 patients who underwent a standardized preoperative 3T MRI

protocol between 2015 and 2021. Institutional review board approval and an exemption from informed consent were granted by the UCSF IRB.

All MRI examinations were acquired on a 3T Discovery 750 scanner (GE Healthcare) equipped with an eight-channel head coil (Invivo). The imaging protocol included 3D T2-weighted imaging, fluid-attenuated inversion recovery (FLAIR), pre- and post-contrast T1-weighted imaging, susceptibility-weighted imaging, DWI, 2D high-angular-resolution diffusion imaging (HARDI), and 3D ASL. As described in prior work (Calabrese et al., 2020, 2021), all images were registered and resampled to a 1-mm isotropic space using Advanced Normalization Tools, with T2-FLAIR serving as the reference volume. Skull stripping was performed using standardized pipelines validated in these earlier studies.

For diffusion imaging, two distinct inputs were considered. The DWI modality corresponds to the *b0* image derived from the diffusion acquisition, while ADC maps were computed separately using standard mono-exponential fitting. These were treated as independent input modalities in the deep learning models.

The dataset includes molecular annotations such as IDH mutation status and MGMT promoter methylation status. Among the 501 cases, 403 (80%) were WHO grade IV gliomas, of which 63% exhibited MGMT promoter methylation. IDH mutation testing was performed using Sanger or next-generation sequencing (Kline et al., 2017), and MGMT methylation status was assessed using a quantitative methylation-specific PCR assay (Kitange et al., 2009), with two or more methylated loci considered positive (Ratai et al., 2018).

For this study, all IDH-wildtype glioblastoma patients (n = 351) were included, comprising 247 MGMT-methylated and 104 unmethylated cases. Patients were randomly assigned to training and testing sets using an 8:2 ratio. Representative examples of the conventional and advanced MRI sequences used in the analysis are shown in Figure 2.

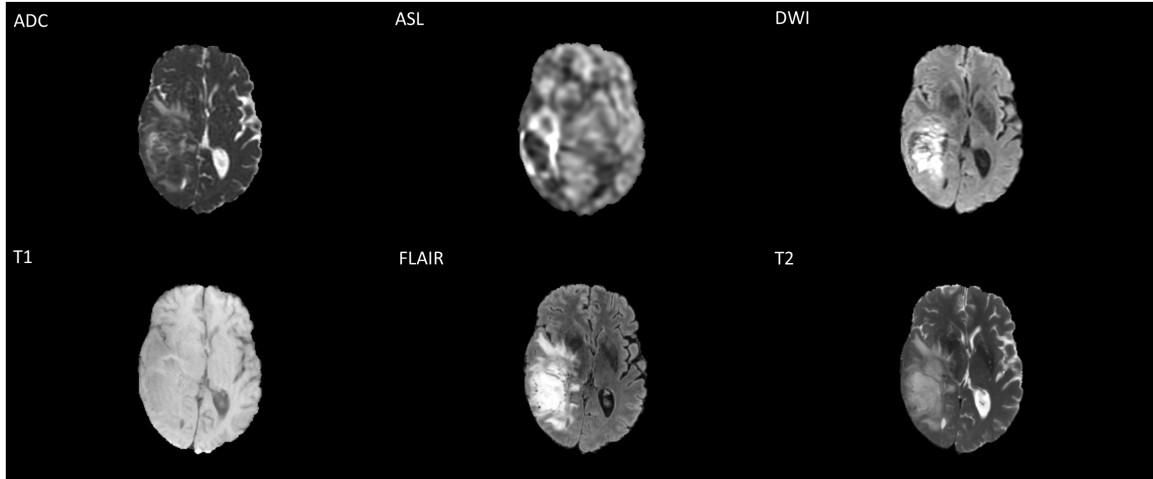

Figure 2: Illustrative example of conventional (T1, T2, FLAIR) and advanced MRI sequences in a GBM patient, including ASL perfusion maps, ADC maps, and diffusion-weighted imaging (DWI, b0 image). The tumor region is visible across modalities.

## 2.2. Data preprocessing

MRI scans were initially converted from NIfTI format to DICOM format using Python to standardize downstream processing. Each scan consisted of 155 axial slices with an in-plane resolution of 240×240 pixels, of which the first 133 slices contained cerebral anatomy. To ensure consistent anatomical coverage across subjects, the central 100 slices were selected for model input.

Formally, an MRI volume can be represented as an ordered set of slices:

$$V = \{v_1, v_2, \ldots, v_{155}\}.$$

Assuming the cerebral region spans slices 1–133, the central subset is extracted as:

$$V_{\text{central}} = \{v_{28}, \ldots, v_{127}\},$$

yielding a fixed-length input of 100 slices per subject.

All selected slices were resized to 224×224 pixels to match the ResNet-10 input specification. Data augmentation consisted of four rotational transformations (0°, 90° clockwise, 90° counterclockwise, and 180°) to improve robustness to spatial variability. Image intensities were standardized using min–max normalization. For single-sequence models, the 100 slices were stacked into a single 3D tensor, whereas dual-sequence models concatenated corresponding MRI sequences along the channel dimension to form a two-channel 3D input.

ASL images followed the standardized UCSF-PDGM preprocessing pipeline, including registration, resampling, skull stripping, and intensity normalization. No additional retrospective motion correction was applied beyond this pipeline; however, cases with gross image corruption or severe artifacts were excluded at the dataset level. Data augmentation further contributed to robustness against residual noise and inter-scan variability.

## 2.3. Deep learning model implementation

MGMT promoter methylation prediction was formulated as a binary classification task. Each MRI input volume was represented as a 3D tensor:

$$X \in \mathbb{R}^{C \times H \times W \times S},$$

where $C$ is the number of MRI channels (1 for single-sequence, 2 for fused ASL+ADC inputs), and $S$ is the number of axial slices. The 3D ResNet-10 model $f_\theta$ produced a methylation probability:

$$\hat{y} = \sigma(f_\theta(X)),$$

and network parameters were optimized using binary cross-entropy:

$$\mathcal{L} = -\left[y \log(\hat{y}) + (1 - y) \log(1 - \hat{y})\right], \quad y \in \{0, 1\}.$$

**Architecture.** All models were built using a 3D ResNet-10 backbone consisting of:

- an initial 7×7×7 3D convolution (64 output feature channels, stride 1, padding 3),

- BatchNorm3d and ReLU activation,

- a 3×3×3 max-pooling layer (stride 2),

- four residual blocks with hierarchical downsampling (stride 2 where appropriate), and

- an AdaptiveAvgPool3d layer yielding a global feature vector before the final fully connected classifier.

Each residual block computes:

$$\mathbf{z}_{l+1} = \mathbf{z}_l + F(\mathbf{z}_l; \theta_l),$$

where $F$ denotes a two-layer 3D convolutional transformation. This skip-connection structure facilitated stable training of deep 3D networks. A batch size of 1 was used due to the high memory demand of 3D volumetric inputs.

**Single-sequence and fused models.** Separate models were trained using ASL, ADC, and DWI inputs. For dual-sequence fusion, ASL and ADC volumes:

$$X^{(\mathrm{ASL})},\ X^{(\mathrm{ADC})} \in \mathbb{R}^{H \times W \times S}$$

were concatenated channel-wise:

$$X_{\mathrm{fusion}} = \mathrm{Concat}\left(X^{(\mathrm{ASL})},\ X^{(\mathrm{ADC})}\right) \in \mathbb{R}^{2 \times H \times W \times S}.$$

This fused representation was processed by a shared 3D ResNet-10 encoder. A T2_FLAIR model served as a conventional MRI baseline.

**Training protocol.** Training was performed using stratified 5-fold cross-validation for 15 epochs per fold. The Adam optimizer was used with an initial learning rate of $1 \times 10^{-4}$, reduced to $5 \times 10^{-5}$ during training.

## 2.4. Statistical analysis

Performance was quantified using accuracy, recall, precision, and F1 score. Overall discriminative performance was evaluated using the area under the receiver operating characteristic curve (AUC). The DeLong test was used to assess statistical differences between the dual-sequence model and single-sequence baselines. Group comparisons for age were performed using the Mann–Whitney U test, and categorical variables (sex and MGMT status) were analyzed using Fisher's exact test. A significance threshold of $p < 0.05$ was applied.

All preprocessing and model development were performed using Python 3.10.12 on a high-performance computing workstation equipped with an NVIDIA A100-SXM4-40GB GPU, an Intel® Xeon® 2.20GHz CPU, 83.5 GB RAM, and 201.2 GB storage.

## 3. Results

### 3.1. Patient characteristics

The demographic and clinical characteristics of the study cohort are summarized in Table 1. There were no significant differences between the training (n = 281) and testing (n = 70) sets in terms of age, sex distribution, or MGMT promoter methylation status. The mean age was 61.43 ± 11.28 years in the training set and 63.53 ± 10.63 years in the testing set

(Mann–Whitney U, $p = 0.253$). The proportion of male patients was similar across subsets (59.1% vs. 65.7%; Fisher's exact test, $p > 0.05$). The prevalence of MGMT promoter methylation was also comparable (70.46% vs. 70.00%; $p > 0.05$), confirming balanced distributions of key covariates following random split.

Table 1: Demographic and clinical characteristics of the study population.

| Parameter | Training set (n=281) | Testing set (n=70) | p-value |
|---|---|---|---|
| Age (mean $\pm$ SD) | 61.43 $\pm$ 11.28 | 63.53 $\pm$ 10.63 | 0.2530 |
| Gender | | | |
| Male | 166 (59.07) | 46 (65.71) | 0.3095 |
| Female | 115 (40.93) | 24 (34.29) | |
| MGMT status | | | |
| Methylated (%) | 198 (70.46) | 49 (70.00) | 0.9395 |
| Unmethylated (%) | 83 (29.54) | 21 (30.00) | |

### 3.2. Performance of single-sequence models

Performance metrics for models trained on individual MRI sequences are presented in Table 2. Among the advanced MRI modalities, the ASL-based model achieved the strongest results, with an accuracy of 0.7571, recall of 0.6361, precision of 0.7333, and an F1 score of 0.6814. The ADC model yielded comparable accuracy (0.7559) but slightly lower recall (0.6062) and F1 score (0.6093). The DWI model demonstrated reduced performance overall, with an accuracy of 0.7042 and an F1 score of 0.5834.

Table 2: Performance of single-sequence models for predicting MGMT promoter methylation.

| Model | Accuracy | Recall | Precision | F1 score |
|---|---|---|---|---|
| ASL | **0.7571** | **0.6361** | **0.7333** | **0.6814** |
| DWI | 0.7042 | 0.5552 | 0.6151 | 0.5834 |
| ADC | 0.7559 | 0.6062 | 0.6129 | 0.6093 |
| T2_FLAIR | 0.6857 | 0.5442 | 0.5829 | 0.5628 |

Best-performing results for each metric are highlighted in bold.

The model based on conventional T2_FLAIR sequences exhibited the lowest predictive performance, with metrics ranging from 0.5442 to 0.6857. These results highlight that advanced MRI sequences, particularly ASL and ADC, provide more informative features for MGMT promoter methylation prediction than structural imaging alone.

### 3.3. Performance of combined ASL+ADC model

The dual-sequence model integrating ASL and ADC inputs achieved the highest overall performance, with an AUC of 0.8163 (Table 3). This represented a statistically signifi-

cant improvement over both the single-sequence ASL model (AUC 0.7580; DeLong test, $p < 0.0001$) and the conventional T2_FLAIR model (AUC 0.6524; $p = 0.0203$). The corresponding receiver operating characteristic (ROC) curves for the combined and baseline models are shown in Figure 3.

For the combined ASL+ADC model, AUC was selected as the primary evaluation metric because it provides a threshold-independent assessment of discriminative performance, which is particularly appropriate for MGMT promoter methylation prediction, where optimal clinical decision thresholds remain uncertain.

Table 3: Comparison of AUC values for the combined ASL+ADC model and baseline models.

| Model     | ASL      | Combined | T2_FLAIR |
|-----------|----------|----------|----------|
| AUC score | 0.7580   | 0.8163   | 0.6254   |
| p-value   | < 0.0001 |          | 0.0203   |

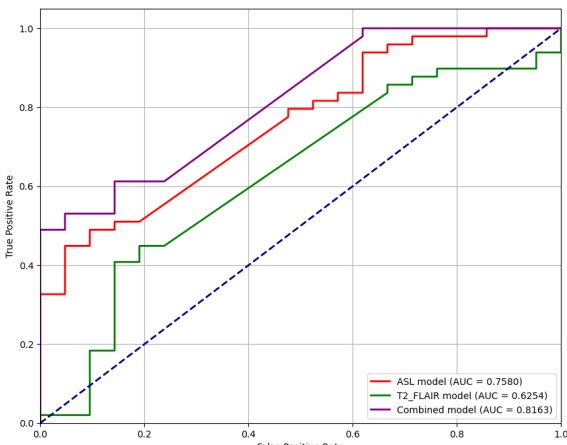

Figure 3: Receiver operating characteristic (ROC) curves for MGMT methylation prediction models. Comparison of ROC curves for the combined ASL+ADC model, the ASL-only model, and the T2_FLAIR baseline, demonstrating the improved discriminative performance of integrating advanced MRI sequences.

These results indicate that integrating perfusion- and diffusion-based physiological information yields a complementary representation that enhances the non-invasive prediction of MGMT promoter methylation status in GBM.

## 4. Discussions

The increasing availability of multimodal neuroimaging datasets such as UCSF-PDGM (Calabrese et al., 2022) has accelerated research on MRI-based artificial intelligence (AI) approaches for characterizing glioblastoma. Leveraging this resource, the present study demonstrates that advanced physiological MRI sequences (particularly ASL and ADC) carry complementary information that improves non-invasive prediction of MGMT promoter methylation status, an important biomarker for treatment planning and prognosis in GBM.

Prior studies have explored a range of imaging planes and acquisition strategies for MRI-based glioma characterization (Batra et al., 1988; Ren et al., 2021; Ding et al., 2021). Although 3D volumetric analysis provides rich contextual information, axial slices remain the most clinically interpretable and are commonly used in radiological workflows (Koh et al., 2010). Our slice-based 3D CNN approach aligns with this clinical convention while enabling efficient model training. The selection of ResNet-10 was motivated by its favorable balance between representational capacity and computational complexity (Faghani et al., 2023; Saxena et al., 2023; Korfiatis et al., 2017; Saeed et al., 2023), making it suitable for high-dimensional MRI inputs without requiring extensive model tuning.

Among the advanced MRI modalities evaluated, ASL and ADC emerged as the most informative single-sequence predictors, a finding that can be explained by their underlying physiological characteristics. ASL provides direct measurements of tissue perfusion, which may reflect tumor vascularity and hypoxia—features known to differ between methylated and unmethylated MGMT phenotypes. In contrast, ADC captures tissue diffusivity and microstructural integrity, reflecting variations in cellularity and necrosis. As these modalities probe distinct but complementary biological processes, ASL and ADC were selected for combination and achieved the strongest individual performance among all evaluated sequences. Other combinations, such as T2+ASL or T2+ADC, were not prioritized due to limited incremental gains observed in preliminary analyses and space constraints inherent to the short-paper format. By integrating perfusion- and diffusion-based information, the combined ASL+ADC model achieved a significantly higher AUC (0.8163) than either modality alone and outperformed a conventional T2_FLAIR–based model, suggesting that perfusion-diffusion fusion provides a richer characterization of tumor physiology relevant to MGMT methylation.

Our findings are consistent with earlier studies that reported limited performance when relying solely on conventional MRI for MGMT prediction. For example, Yogananda et al. achieved an AUC of 0.6588 using T2-weighted imaging with a 3D Dense-UNet (Yogananda et al., 2021), while Saxena et al. reported an AUC of 0.753 using 3D ResNet-18 (Saxena et al., 2023). By incorporating advanced physiological sequences, our ASL and combined models exceed these benchmarks, reinforcing the value of moving beyond purely structural imaging for molecular phenotype estimation.

Despite these encouraging results, several limitations should be acknowledged. First, this study relied on a single model architecture and a single-center cohort; exploring alternative architectures (e.g., DenseNet, EfficientNet, or transformer-based 3D models) and validating performance on external datasets will be important for assessing generalizability. However, public GBM datasets that provide both ASL imaging and matched MGMT promoter methylation labels remain scarce. Second, although ASL and ADC provided strong

and complementary predictive signals, ASL is not universally included in standard brain tumor imaging protocols, and other advanced sequences (e.g., perfusion-weighted imaging or spectroscopy) were not evaluated and may contribute additional information where available. Third, no clinical variables or radiomic features were incorporated; prior studies suggest that multimodal fusion of imaging and clinical data may yield more holistic and robust predictors (Huang et al., 2020). Finally, this work focused on axial-plane imaging without voxel-level explainability analyses; future studies incorporating cross-plane fusion and interpretability methods such as Grad-CAM may further enhance robustness and clinical interpretability.

Together, these findings suggest that physiologically informed MRI sequences capture molecularly relevant tumor characteristics that are not fully reflected in conventional structural imaging. The complementary contributions of perfusion- and diffusion-based signals highlight the importance of integrating functional imaging into deep learning–based molecular profiling. As advanced MRI acquisition becomes increasingly common in neuro-oncology, such approaches may improve clinical trust in non-invasive biomarkers and facilitate more informed treatment stratification in GBM.

## 5. Conclusion

This study evaluated advanced physiological MRI sequences for non-invasive prediction of MGMT promoter methylation status in IDH-wildtype glioblastoma and demonstrated that ASL and ADC offer complementary information that substantially improves predictive performance. The combined ASL+ADC 3D CNN model achieved the highest accuracy and discriminative ability, outperforming both single-sequence models and a conventional T2_FLAIR baseline. These findings highlight the value of integrating perfusion- and diffusion-based imaging for molecular phenotype estimation and support the potential clinical utility of advanced MRI–based deep learning tools in guiding treatment planning for GBM.

Future work should focus on improving generalizability through external validation and multimodal integration, including clinical metadata and multi-plane imaging. Exploring more diverse model architectures and incorporating explainability mechanisms may further enhance robustness, interpretability, and clinical applicability in precision neuro-oncology.

## Acknowledgments

This work is supported by the National Science and Technology Council, Taiwan [grant number NSTC114-2221-E-038-015].

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
