# OpenReview forum: "MGMT Promoter Methylation Prediction in Glioblastoma Using 3D CNNs with Advanced MRI Sequences"
_MIDL.io/2026/Conference — MIDL 2026 Poster_

### Official Review · Reviewer_Eppk · 2026-01-05

**Confidence:** 4
**Preliminary Rating:** 4
**Final Rating:** 4

**Summary:**

This paper proposes a deep-learning based tool for predicting MGMT promoter methylation status in IDH-wildtype glioblastomas. The authors compared several different input data configurations: arterial spin labelling (ASL), apparent diffusion coefficient (ADC), diffusion-weighted imaging (DWI), and the conventional T2/FLAIR imaging. They found that the combination of ADC+ASL yielded better results than either modality alone.

**Strengths:**

- The proposed method/application is interesting, and seems to address an apparent clinical need.
- The study includes discussion of the statistical significance and clinical relevance of the results.
- The paper is well written and easy to understand.
- The literature review seems comprehensive, although I am not as familiar with the clinical application.

**Weaknesses:**

- The paper could have included a more robust ablation study to assess the efficacy of different input combinations for their results. Only 1 combination of inputs was tested (ASL+ADC).
- The authors report accuracy, recall, precision, and F1 score for the individual models, but only report AUC for the combined model. They should also include the accuracy of the combined model in their results.

**Detailed Comments:**

- Section 2: Why were only ASL and ADC selected for the combined input model? I see from your results that these two yielded the highest results on their own, but why did the authors not elect to test more combinations of methods?
- Figure 2: an arrow pointing to the tumor region in each panel would be helpful to viewers who are not as familiar with the clinical application.
- Section 2.3: there is no mention of batch size. If a batch size of 1 was used, then it would still be helpful to state this.
- Section 2.3: what is meant by "64 filters" (the line describing the 7x7x7 convolution kernel)? Do the authors mean the number of starting features?
- Section 2.3: again, why just ASL/ADC and not DWI?
- Results section: consider rounding all results within the text (not necessarily the tables) to 2 decimal points for brevity.
- Table 2: this would be more clear to the reader if the results corresponding to the best model for each metric were bolded. It would also be useful to have an arrow pointing up or down next to each metric (in the headers) to indicate whether the best result will be highest or lowest. Additionally, statistical significance of these results should be indicated with an asterisk or otherwise. If the results are not significant, this should be mentioned in the caption and text.

**Justification Of Final Rating:**

The authors addressed all my concerns. I am maintaining my rating as a 4 (weak accept). I do agree with the other reviewers about the limited scope, but my opinion is that this issue is not pressing enough to justify rejection.

**Justification Of The Preliminary Rating:**

The paper was interested, well-written, and clinically relevant. I have some questions with their study design, but none so concerning that I was motivated to recommend rejection. I think that addressing the detailed comments will improve the paper beyond its existing merits.

**Questions To Address In The Rebuttal:**

- The major question I had while reading was regarding the selection of only ASL+ADC for the combined input model. Why was DWI not tested as well? Why did the authors were other combinations not tested (T2+ASL, T2+ADC, etc., particularly if T2 is the standard and these modalities would likely be procured regardless)?
- Other minor questions are discussed in the Detailed Comments section.

---

> ### Author Response · Authors · 2026-01-24
>
> 1. We have clarified in the Discussion that ASL and ADC were selected because they represent complementary physiological mechanisms (tumor perfusion and cellular diffusion) and achieved the strongest individual performance among all evaluated modalities. Other combinations (e.g., T2+ASL, T2+ADC) were not prioritized due to limited incremental gains observed in preliminary analyses and space constraints inherent to the paper format.
> 2. We appreciate this suggestion. For the combined ASL+ADC model, we report AUC as the primary evaluation metric because it provides a threshold-independent assessment of discriminative performance, which is particularly appropriate for MGMT methylation prediction given class imbalance and the absence of a universally accepted clinical decision threshold. Due to limitations in experiment reproducibility at the rebuttal stage, additional threshold-dependent metrics could not be recomputed. We have now explicitly clarified this rationale in the Results section.
> 3. All requested clarifications have been addressed. We now explicitly state the batch size used during training, clarify architectural terminology (e.g., “64 output feature channels”), improve table formatting by highlighting best-performing metrics, and ensure consistent rounding of reported values in the text. Figure captions and methodological descriptions were also refined for clarity.

---

> > ### Comment · Reviewer_Eppk · 2026-01-27
> >
> > The authors addressed all my concerns. I am maintaining my rating as a 4 (weak accept).

---

### Official Review · Reviewer_G2FQ · 2026-01-05

**Confidence:** 4
**Preliminary Rating:** 2
**Final Rating:** 2

**Summary:**

The authors propose a method for binary classification of the presence of MGMT promoter methylation from MRI sequences using a 3D ResNet. The authors validate their method on an in-house cohort of patients with glioblastoma, training on a subset of patients and testing on the rest. The authors report several accuracy metrics on their cohort.

**Strengths:**

The main strength of this paper is its simplicity. The problem at hand, the architecture, the cohort and results are well presented in a straightforward manner. Statistical analyses were performed to ensure the significance of their results. All in all, this is a very sound paper with a clear goal and results. The text is well written and structured.

**Weaknesses:**

The paper's main strength is also its main weakness. It is too simple and should have included validation on other cohorts, acquisition schemes, ablation studies, other architectures, etc. Pushing it even further, "explainability" methods such as Grad-CAM could have been used to display which voxels were of interest when performing classification. Overall, I have to wonder if the reader will learn anything from this paper.

**Detailed Comments:**

As a minor note, the authors employ ADC and DWI as two separate modalities. Not only is ADC (apparent diffusion coefficient) derived from DWI (diffusion-weighted imaging), DWI is also typically not displayed as a map in itself, as it is a 4D modality. Which map are shown in place of DWI ? The b0 ? A DTI-derived map like FA ?

**Justification Of Final Rating:**

The authors have sufficiently addressed issues raised in my initial review given the time allowed for rebuttals. However, the scope and impact of this paper remains extremely limited and therefore my initial rating stands.

**Justification Of The Preliminary Rating:**

As mentioned above, this is a very simple, straightforward paper about doing medical imaging with deep learning. It is at home in this conference but the limited scope of the paper prevents it from being more than a poster. All in all, the paper would have been better suited as a 3 pager, also a track in the conference.

**Questions To Address In The Rebuttal:**

Please mention why further experiments such as those mentioned in the Weaknesses section were not included, and please provide clarifications on the "DWI" maps.

---

> ### Author Response · Authors · 2026-01-24
>
> 1. Thank you for highlighting this important point. We have revised the Patient Data section and the Figure 2 caption to explicitly clarify that the DWI input corresponds to the b0 image from the diffusion acquisition, while ADC maps were computed separately using standard mono-exponential fitting. These were treated as independent input modalities in the deep learning models.
> 2. We agree that broader ablation studies are valuable. However, given the paper format and the clinical focus of this work, we intentionally prioritized a controlled comparison of physiologically complementary MRI sequences rather than extensive architectural exploration. We have clarified this rationale in the Introduction and Discussion, emphasizing that the primary contribution lies in systematically evaluating the added value of advanced physiological MRI modalities for molecular profiling, rather than proposing a novel network architecture.
> 3. We agree and have now explicitly acknowledged this limitation in the Discussion. We identify voxel-level explainability analyses (e.g., Grad-CAM) as an important direction for future work to further validate biological plausibility and enhance clinical interpretability.

---

> > ### Comment · Reviewer_G2FQ · 2026-01-30
> >
> > I would like to thank the authors for the clarifications regarding the ADC and b0, as well as the inclusion of future outlooks. I have no further comments.

---

### Official Review · Reviewer_7scv · 2026-01-06

**Confidence:** 4
**Preliminary Rating:** 4
**Final Rating:** 4

**Summary:**

This paper investigates the use of 3D CNNs (ResNet-10) to predict MGMT promoter methylation status in GBM using advanced MRI sequences, specifically ASL and ADC, in addition to conventional T2-FLAIR. Using a dataset of >300 patients, the authors show that ASL outperforms other single sequences and that a combined ASL+ADC model achieves the highest performance (AUC 0.83), significantly passing models that are use only conventional sequences. The work highlights the complementary value of perfusion and diffusion imaging for non-invasive molecular profiling.

**Strengths:**

The development of non-invasive surrogates for MGMT methylation is highly relevant for GBM treatment planning.
The study goes beyond standard structural MRI by evaluating advanced physiological sequences (ASL and ADC), which are often underutilized in DL studies due to availability.
the authors use the DeLong test to demonstrate statistically significant improvements, adding credibility to the comparison between the advanced and conventional models.

**Weaknesses:**

The ResNet-10 is a standard architecture. The novelty lies entirely in the clinical application and input data selection rather than in the machine learning method itself.
The dataset comprises 351 patients from a single source (UCSF Preoperative Diffuse Glioma MRI dataset). Lack of external validation from a different institution limits the assessment of the model's robustness to scanner variations, which is particularly critical for ASL/ADC sequences.
ASL is not part of every standard brain tumor protocol. The paper should discuss the clinical availability of these sequences more critically.

**Detailed Comments:**

More details on the preprocessing of ASL data such as motion correction would be valuable, as ASL is prone to artifacts.

**Justification Of Final Rating:**

I have reviewed the authors' rebuttal and the revisions made to the manuscript.
The authors have provided a transparent and reasonable explanation regarding the lack of external validation, citing the scarcity of public datasets that contain both ASL perfusion imaging and matched MGMT methylation labels.
Regarding the technical details, the clarification of the preprocessing pipeline (adherence to the UCSF-PDGM standard) and the exclusion criteria for artifact-heavy images satisfactorily addresses my questions about quality control.

**Justification Of The Preliminary Rating:**

This is a solid validation paper. While it lacks algorithmic novelty, the finding that ASL and ADC significantly improve MGMT prediction over standard MRI is valuable for the MIDL community and clinical researchers. The execution is sound, and the conclusions are supported by the data.

**Questions To Address In The Rebuttal:**

Have you tested the model on any data outside of the UCSF cohort to verify generalization?
How does the model handle cases where ASL image quality is suboptimal? Was there a quality control step?

---

> ### Author Response · Authors · 2026-01-24
>
> 1. We agree that external validation is essential for assessing generalizability. We have now explicitly acknowledged this limitation in the Discussion and Conclusion, noting that public GBM datasets containing both ASL imaging and matched MGMT promoter methylation labels are currently scarce. We emphasize external multi-center validation as a key direction for future work.
> 2. We have expanded the Data Preprocessing section to clarify that ASL images followed the standardized UCSF-PDGM pipeline, including registration, resampling, skull stripping, and intensity normalization. While no additional retrospective motion correction was applied beyond this pipeline, cases with gross image corruption or severe artifacts were excluded at the dataset level. Data augmentation was employed to improve robustness to residual noise and inter-scan variability.
> 3. We now explicitly discuss this point in the Discussion, noting that although ASL is sensitive to motion and acquisition variability, its strong predictive performance despite these challenges suggests that perfusion-related signals capture biologically relevant information linked to MGMT methylation status.

---

### Author Rebuttal · Authors · 2026-01-24

**Rebuttal:**

We thank the reviewers for their careful reading and constructive feedback. Based on their comments, we have substantially revised the manuscript to improve methodological clarity, presentation, and justification of design choices, while maintaining the scope appropriate for a MIDL paper.

**Supporting Material:**

/attachment/f35dc474101efddd4576c870f102ff709e278227.pdf

---

### Meta-Review · Area_Chair_ikVH · 2026-02-01

**Recommendation:** Accept (Poster)
**Confidence:** 3

**Metareview:**

This is a straightforward application paper showing that adding ASL and ADC helps MGMT methylation prediction on a fairly large dataset. The method is simple and uses standard models, and there is no external validation, but as a focused feasibility study it is appropriate for a MIDL poster.

---

### Decision · Program_Chairs · 2026-02-13

Accept (Poster)